# Biochemical characterization of protease activity of Nsp3 from SARS-CoV-2 and its inhibition by nanobodies

Lee A. Armstrong[1], Sven M. Lange[1], Virginia Dee Cesare[1], Stephen P. Matthews[1], Raja Sekhar Nirujogi[1], Isobel Cole[1], Anthony Hope[2†], Fraser Cunningham[2], Rachel Toth[3], Rukmini Mukherjee[4,5], Denisa Bojkova[6], Franz Gruber[7], David Gray[2], Paul G. Wyatt[2], Jindrich Cinatl[6], Ivan Dikic[4,5,8], Paul Davies[1], Yogesh Kulathu[1] *

1 MRC Protein Phosphorylation and Ubiquitylation Unit, Sir James Black Centre, School of Life Sciences, University of Dundee, Dundee, Scotland, United Kingdom, 2 Drug Discovery Unit, School of Life Sciences, University of Dundee, Dundee, Scotland, United Kingdom, 3 MRC Reagents and Services, University of Dundee, Dundee, Scotland, United Kingdom, 4 Institute of Biochemistry II, Goethe University Frankfurt Medical Faculty, University Hospital, Frankfurt am Main, Germany, 5 Buchmann Institute for Molecular Life Sciences, Goethe University Frankfurt, Frankfurt am Main, Germany, 6 Institute of Medical Virology, University Hospital Frankfurt, Frankfurt am Main, Germany, 7 National Phenotypic Screening Centre, University of Dundee, Dundee, Scotland, United Kingdom, 8 Max Planck Institute of Biophysics, Frankfurt am Main, Germany

☯ These authors contributed equally to this work.
† Deceased.
* y.kulathu@dundee.ac.uk

**Data Availability Statement:** All proteomics data have been deposited with PRIDE. All cDNA constructs used in the study along with sequence details are available from MRC reagents and

## Abstract

Of the 16 non-structural proteins (Nsps) encoded by SARS CoV-2, Nsp3 is the largest and plays important roles in the viral life cycle. Being a large, multidomain, transmembrane protein, Nsp3 has been the most challenging Nsp to characterize. Encoded within Nsp3 is the papain-like protease domain (PLpro) that cleaves not only the viral polypeptide but also K48-linked polyubiquitin and the ubiquitin-like modifier, ISG15, from host cell proteins. We here compare the interactors of PLpro and Nsp3 and find a largely overlapping interactome. Intriguingly, we find that near full length Nsp3 is a more active protease compared to the minimal catalytic domain of PLpro. Using a MALDI-TOF based assay, we screen 1971 approved clinical compounds and identify five compounds that inhibit PLpro with $IC_{50}$s in the low micromolar range but showed cross reactivity with other human deubiquitinases and had no significant antiviral activity in cellular SARS-CoV-2 infection assays. We therefore looked for alternative methods to block PLpro activity and engineered competitive nanobodies that bind to PLpro at the substrate binding site with nanomolar affinity thus inhibiting the enzyme. Our work highlights the importance of studying Nsp3 and provides tools and valuable insights to investigate Nsp3 biology during the viral infection cycle.

services. All other relevant data are within the manuscript and Supporting information files The original uncropped gel images have been deposited (http://dx.doi.org/10.17632/6ymh4jrxvb.1).

**Funding:** UKRI | Medical Research Council (MRC): Lee A Armstrong,Sven M Lange,Virginia de Cesare, Stephen P Matthews,Raja Sekar Nirujogi,Isobel Cole,Rachel Toth,Paul Davies,Yogesh Kulathu MC_UU_00018/3; EC | H2020 | H2020 Priority Excellent Science | H2020 European Research Council (ERC):Lee A Armstrong,Sven M Lange, Stephen P Matthews,Yogesh Kulathu 677623; Tenovus:Lee A Armstrong,Sven M Lange,Stephen P Matthews,Yogesh Kulathu; UKRI | MRC | Medical Research Foundation:Anthony Hope,Fraser Cunningham,David Gray,Paul G. Wyatt MC_PC_18044; Lister Institute of Preventive Medicine:Yogesh Kulathu; EC | H2020 | H2020 Priority Excellent Science | H2020 European Research Council (ERC):Rukmini Mukherjee,Ivan Dikic 742720.

**Competing interests:** The authors have declared that no competing interests exist.

## Introduction

COVID-19 has taken a significant toll on human life and has impacted society in unprecedented ways. Therefore, understanding the mechanisms by which SARS-CoV-2 interacts with human cells and the functions of the different viral proteins in establishing infection is of paramount importance. The genome of SARS-CoV-2 encodes 16 non-structural proteins (Nsps) which play different roles in the viral life cycle [1]. Nsp3, the largest of the Nsps, is a key component of the viral replication and transcription complex assembled on host cell membranes where replication and transcription of the viral genome occurs [2–4]. Another essential role of Nsp3 is its function as a protease. Encoded within Nsp3 is the papain-like protease (PLpro$^{CoV-2}$) that cleaves between Nsp1-Nsp2, Nsp2-Nsp3 and Nsp3-Nsp4 to release Nsp1, Nsp2 and Nsp3 from the viral polypeptide. While several coronaviruses express two PLpro enzymes, SARS-CoV-2 only expresses one PLpro enzyme [1, 5].

PLpro$^{CoV-2}$ shows many similarities to the PLpro encoded by SARS-CoV-1, PLpro$^{CoV-1}$. The structure of PLpro is similar to that of human ubiquitin specific protease (USP) deubiquitinating enzymes (DUBs), which resembles an extended or open hand with thumb, palm and fingers subdomains that mediate contacts with substrate to form the S1 site [6–9]. In addition to processing viral polypeptides, PLpro possesses deubiquitylating and deISGylating activities as it can also efficiently cleave ubiquitin and ISG15 modifications [10]. Ubiquitylation and ISGylation, the covalent attachment of ubiquitin and the ubiquitin-like modifier (Ubl) ISG15, respectively, to proteins, are important posttranslational modifications (PTMs) in host innate immune antiviral and inflammatory signalling pathways [11]. K48-linked polyubiquitination of proteins, for instance, acts as a signal for proteasomal degradation of the targeted protein [12]. ISG15 is an interferon-stimulated gene whose expression is induced following viral infection and is important to host antiviral responses [13]. ISGylation plays a variety of roles in impeding viral infection. ISGylation of viral capsid proteins can disrupt particle assembly [14], whilst ISGylation of cellular factors can interfere with both assembly and budding of viruses including HIV and influenza [15, 16]. PLpro$^{CoV-1}$ has a unique mode of action as it recognizes ubiquitin at an additional S2 ubiquitin-binding site to bind and cleave K48-linked ubiquitin chains from substrates [17, 18]. ISG15 contains two connected Ubl domains which together resemble diubiquitin that also bind to the S1 and S2 sites on PLpro. These additional protease activities of PLpro antagonize host ubiquitylation and ISGylation to effectively subvert antiviral signalling and the host immune response. These multifaceted roles of PLpro render it essential for viral replication, thereby making it an attractive therapeutic target for COVID-19.

Given the large size of Nsp3, PLpro and the other domains in Nsp3 have each been individually characterized in detail. However, most studies have examined only the minimal proteolytic PLpro domain and there is little known about the influence of the other domains in Nsp3 on protease activity and function. In the present study, we perform detailed biochemical characterization of minimal PLpro$^{CoV-2}$ and compare it with the protease activity of Nsp3$^{CoV-2}$. We make the surprising observation that extended Nsp3 is a more active protease cleaving both ISG15 and K48-linked chains with greater efficiency. Intriguingly, while Nsp3 can very efficiently cleave the viral polypeptide Nsp1-Nsp2, the minimal PLpro is unable to cleave this fusion. To find inhibitors of PLpro, we employed a high-throughput MALDI TOF-based assay to screen 1971 FDA approved compounds and identify inhibitors of the protease with IC$_{50}$ values in the 1–10 mM range. Despite displaying good inhibition *in vitro*, none of the inhibitors show any efficacy in blocking viral replication in a SARS-CoV-2 viral infection model. We therefore engineered nanobodies that bind to and inhibit PLpro in the nanomolar range. These nanobodies will be valuable tools to identify cellular substrates of Nsp3 and to investigate Nsp3 biology.

## Results and discussion

### Characterization of protease activity of PLpro and Nsp3

SARS-CoV-2 PLpro, PLpro$^{CoV-2}$, is a constituent domain of the 217 kDa multidomain membrane protein, Nsp3 (**Fig 1A**). PLpro is a multifunctional enzyme with three distinct protease activities that catalyse the cleavage of (i) K48-linked polyubiquitin, (ii) ISG15 and (iii) viral polypeptides. Processing of the viral polypeptide involves cleavage between Nsp1-Nsp2, Nsp2-Nsp3 and Nsp3-Nsp4 (**Fig 1A**) [19]. While recent studies have characterised the substrate specificity and cleavage activity of PLpro$^{CoV-2}$ towards polyUb and ISG15 [7, 9, 20, 21], we wondered if elements outside the minimal PLpro domain might play a role in substrate recognition and cleavage. Nsp3 is a multidomain protein containing a transmembrane segment that mediates its localization to membrane vesicles [22, 23]. In addition to the protease domain, Nsp3 contains an N-terminal Ubl (ubiquitin-like domain) followed by an ADRP (ADP-ribose phosphatase, SUD (SARS-unique domain) domain, another Ubl domain that is part of PLpro. C-terminal of the protease is a NAB (nucleic acid binding) domain followed by a TM (transmembrane segment) and a ZnF (zinc finger) domain at the very C-terminus [23, 24]. Despite being a large multidomain protein, we obtained milligram amounts of pure and stable recombinant Nsp3 truncations when we either removed or replaced the transmembrane segment and the acidic stretch at the N-terminus with a linker (**Fig 1B**). As all the Nsp3 variants were equally active (**S1B Fig**), Nsp3 spanning residues 179–1329 (henceforth referred to as Nsp3$^{core}$) was chosen for further characterization.

To understand how PLpro can cleave such structurally distinct substrates and to characterize these three activities further relative to Nsp3, we first generated K48-linked Ub3, pro-ISG15 and, to monitor the cleavage of the viral polypeptide, we expressed and purified full length Nsp1 fused to the first 19 amino acids of Nsp2 (Nsp1-2Δ). We compared the activities of PLpro and Nsp3$^{core}$ for each of the substrates in qualitative gel-based assays. As previously observed for PLpro$^{CoV-1}$, PLpro$^{CoV-2}$ is selective at cleaving K48-linked chains (**S1A Fig**) and shows a similar mode of cleavage by cleaving triubiquitin into diubiquitin and monoubiquitin as final products [7, 9, 18]. In our assay, nearly all triubiquitin is cleaved by Nsp3$^{core}$ (50 nM) after 15 minutes whereas a significant portion is still to be cleaved by PLpro (**Fig 1C, upper**). Similarly, Nsp3$^{core}$ (20 nM) cleaves most of the pro-ISG15 by 15 minutes while only ~50% is cleaved by PLpro (**Fig 1C, lower**). Intriguingly, Nsp3$^{core}$ (1 μM) is able to cleave the Nsp1-2Δ substrate, whilst PLpro is not (**Fig 1D**). To confirm this observation, we incubated increasing concentrations of PLpro with Nsp1-2Δ, which remained resistant to cleavage (**S1C Fig**). We therefore conclude that elements outside the minimal PLpro domain are responsible for recognising the viral polypeptide for cleavage. The Nsp1-2Δ substrate tested so far only contains 19 amino acids from Nsp2. To test if a more physiological form of the viral polypeptide could be cleaved by PLpro, we expressed and purified full-length Nsp1-Nsp2 fusions. We again find that PLpro is unable to cleave Nsp1-Nsp2 FL substrate, whereas Nsp3$^{core}$ readily cleaves it (**Fig 1E**).

These qualitative assays suggest that Nsp3 is a more active enzyme. To confirm these observations, we used a MALDI TOF based DUB assay (Ritorto et al., 2014), which shows that PLpro efficiently cleaves K48 trimers with a $K_m$ of 29.39 μM, a $V_{max}$ of 0.66 mM$^*$min$^{-1}$ and a $k_{cat}$ of 22 min$^{-1}$ (**S1G Fig**). In contrast, Nsp3 cleaves K48-linked triubiquitin with a $K_m$ of 17.79 μM, a $V_{max}$ of 0.62 mM$^*$min$^{-1}$ and a $k_{cat}$ of 20.6 min$^{-1}$. The comparatively lower $K_m$ of Nsp3 potentially explains the higher cleavage activity observed. Even though Nsp3 is more active than PLpro, it still retains its specificity for cleaving K48-linked polyubiquitin and does not cleave any of the other polyubiquitin linkages tested (**Fig 1F**). The C-termini of all PLpro substrates contain a -GG motif, with PLpro cleaving after the last Gly [20] (**S1D Fig**). Indeed,

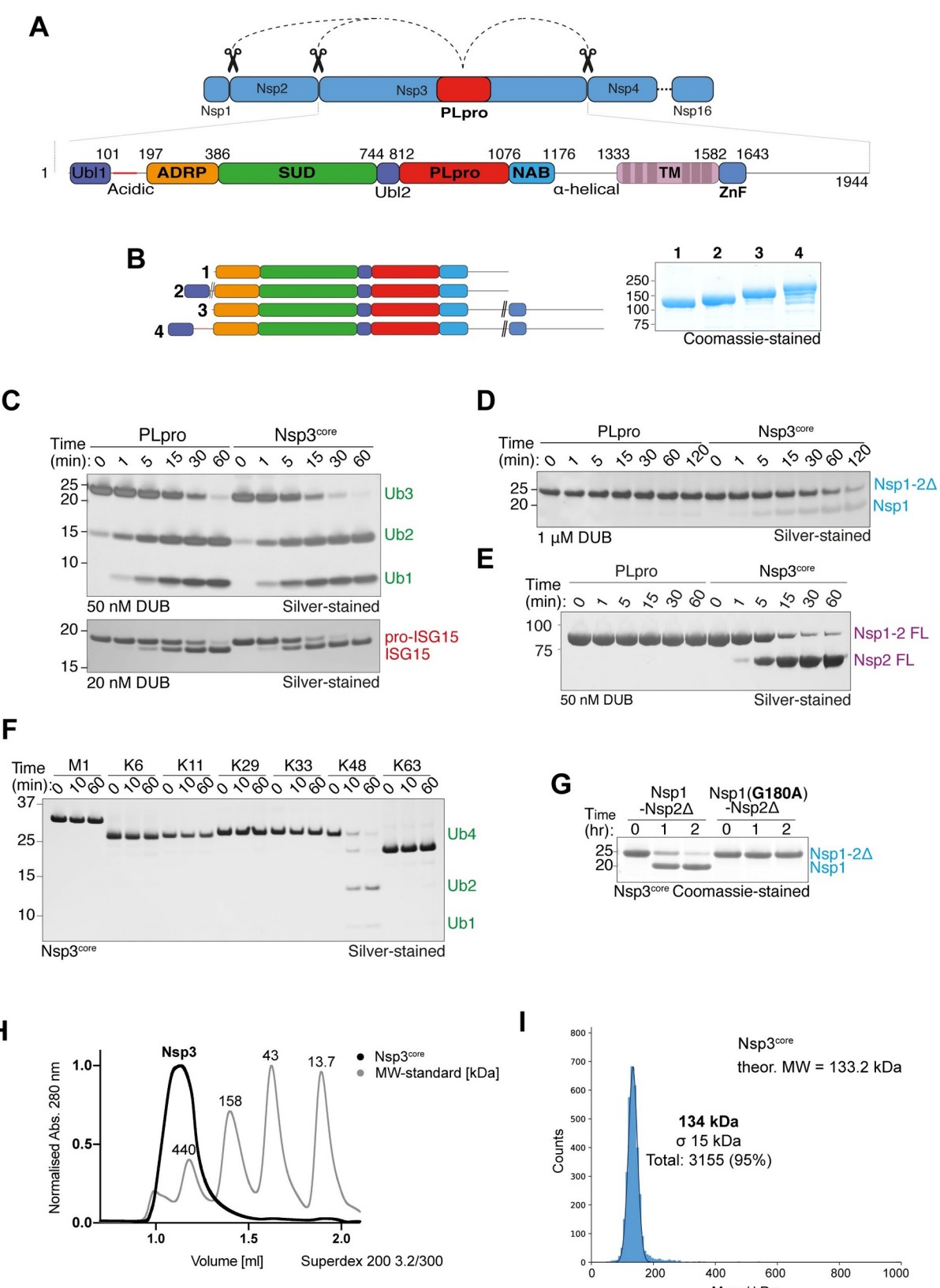

**Fig 1. Nsp3<sup>core</sup> is a more active protease compared to PLpro. A)** Schematic of cleavage of Nsp1-3 by PLpro from the viral polypeptide and the domain structure of Nsp3: Ubl (ubiquitin-like domain), ADRP (ADP-ribose phosphatase, SUD (SARS-unique domain), PLpro (papain-like protease), NAB (nucleic acid binding domain), TM (Transmembrane domain), ZnF (zinc finger motif). **B)** Coomassie-stained gel of Nsp3 constructs used in this study with accompanying domain schematics (colour coding as in 1A; double slash indicates deletion). **C)** DUB assay directly comparing cleavage of Ub3 (upper) and ISG15 (lower) by PLpro and Nsp3<sup>core</sup>. **D)** Assay directly comparing cleavage of Nsp1-2Δ by PLpro and Nsp3<sup>core</sup>. **E)** Protease assay comparing the cleavage of Nsp1-2 FL by PLpro and Nsp3<sup>core</sup>. **F)** DUB assay showing the polyubiquitin linkage preference of Nsp3<sup>core</sup>. **G)** Cleavage assay comparing cleavage of Nsp1-2Δ and Nsp1-2Δ G180A mutant by Nsp3<sup>core</sup>. **H)** UV-traces of analytical gel filtration analyses (Superdex 200 3.2/300) of Nsp3<sup>core</sup> overlaid onto molecular weight standards. **I)** Mass photometry analysis measuring the molecular mass of Nsp3<sup>core</sup>. Data shown are representative of two independent experiments.

mutating the C-terminal G180 of Nsp1 renders it resistant to cleavage by Nsp3 as does mutation of the catalytic C856 of Nsp3 to Ala (**Fig 1G, S1E Fig**), confirming that the extended Nsp3 maintains substrate specificity and does not contain additional protease catalytic sites. We then generated propargylated Nsp1 (Nsp1<sup>Prg</sup>), a reactive probe which would covalently modify the catalytic Cys of PLpro and Nsp3 [25]. Indeed, while Nsp3<sup>core</sup> was efficiently modified by Nsp1<sup>Prg</sup>, barely any conversion was observed for PLpro (**S1F Fig**).

Since Nsp3<sup>core</sup> efficiently cleaves the viral polypeptide fusion Nsp1-Nsp2, which minimal PLpro does not cleave, and Nsp1<sup>core</sup> is readily modified by Nsp1<sup>Prg</sup> leads us to speculate that regions outside of the PLpro domain may mediate additional interactions with Nsp1 and Nsp2 to facilitate cleavage. As Nsp3 is a multidomain protein, it is difficult to predict which of these may play a role in binding to the viral polypeptide. The SUD domain of SARS-CoV-2 shares 75% sequence identity with that of SARS-CoV-1 SUD and consists of various macrodomains (Mac1, Mac2, Mac3 and DPUP) which bind to nucleotides and to proteins such as the E3 ligase RCHY1 and polyA binding protein 2 [23, 24, 26–29]. In addition to these roles, it is possible that the SUD domain may mediate additional interactions with Nsp1 and Nsp2 to promote viral polypeptide cleavage.

Despite several attempts, we were unable to crystallise longer variants of Nsp3 on its own or in complex with a substrate. Intriguingly, Nsp3 elutes at an apparent molecular weight of ~520kDa on a size-exclusion chromatography column raising the possibility that it could be tetramer (**Fig 1J**). However, mass photometry analysis reveals Nsp3 to be a monomer (**Fig 1K**), suggesting that the behaviour on gel filtration could be attributed to an extended conformation or coiled-coil content in Nsp3. Obtaining a structure of Nsp3 in complex with Nsp1-Nsp2 will reveal why it is more active compared to PLpro and the interactions mediated with Nsp1 and Nsp2.

## Interactome of PLpro and Nsp3

Since we observe differences in protease activity between Nsp3<sup>core</sup> and PLpro, we next wanted to compare the interactomes of PLpro and Nsp3. To date most efforts to comprehensively define the interactomes of SARS-CoV-2 non-structural proteins have failed to identify the interactome of Nsp3 [30]. When we expressed full length Nsp3 in HEK 293 cells, we observed very low expression levels that made it challenging to get enough material for affinity enrichment/mass spectrometry (AE/MS) analysis. Hence, we made a truncation where we deleted the transmembrane segments of Nsp3 (Nsp3ΔTM). The lung epithelial cell line A549 was transfected with either Strep-tagged Nsp3ΔTM or PLpro and cells were lysed following IFN-α stimulation for 36 hours.

Quantitative mass spectrometry analysis revealed that Nsp3 ΔTM and PLpro had a largely overlapping interactome (**Fig 2**). Interestingly, many of the interactors enriched for both PLpro and NSP3 DTM are interferon stimulated and proteins involved in antiviral signalling. With its preference for cleaving ISG15, we observed ISG15 as a significant interactor that was also identified in the PLpro<sup>CoV-2</sup> interactome [9]. Other prominent overlapping interactors

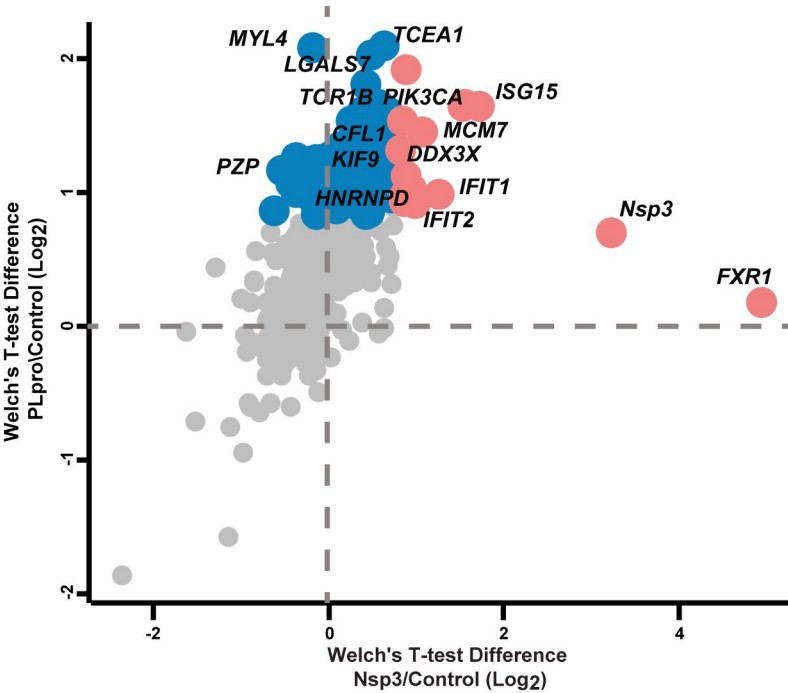

**Fig 2. Comparison of Nsp3 and PLpro interactome.** Volcano plot showing significant interactors identified by pull down on Nsp3 DTM or PLpro from transiently transfected cells followed by quantitative mass spectrometry. Interactors enriched in PLpro pull downs are highlighted in blue and those enriched in Nsp3 pull downs are highlighted in red circles.

include the IFITs (Interferon-induced protein with tetratricopeptide repeats), IFIT1 and IFIT2, antiviral proteins that function as sensors of viral single-stranded RNAs to inhibit expression of viral messenger RNAs [31]. Of note, IFIT1 (also known as p56 or ISG56) has previously been recognised as a target of ISGylation [32]. Nsp3 ΔTM also interacts with the antiviral proteins SAMHD1, which binds to nucleotides and depletes cellular dNTPs, and 1'-5'-oligoadenylate synthase 3 (OAS3), another interferon-induced gene that inhibits viral replication by degrading viral RNA [33]. In addition, both PLpro and Nsp3 ΔTM interact with the multifunctional ATP-dependent RNA helicase DDX3X, which promotes IRF3 activation leading to the production of the type I interferons, IFN-α and IFN-β [34].

Our data also identifies interactors linked to translation (EIF5A2, EIF2S1, ICE1, EIF2S3L) and the actin cytoskeleton (Cofilin-1, tubulins, KIF9, MYL4, PDLIM5), which was also observed in a recent study that used proximity labelling [35]. While most interactors are largely overlapping between Nsp3 ΔTM and PLpro, we identify the RNA binding protein Fragile X mental retardation syndrome-related protein 1 (FXR1) to be enriched only in Nsp3 ΔTM pull downs. Interestingly, Nsp3 from Sindbis virus interacts with FXR1 to bind viral RNAs for the assembly of viral replication complexes [36].

Despite having several additional domains, no additional unique interactors were identified for Nsp3 ΔTM. We attribute the lack of additional interactions to Nsp3 ΔTM to the following possibilities: (i) the Nsp3 used in our experiments lacks the transmembrane segment and hence does not localize to the native subcellular compartment where the relevant cellular interactors of Nsp3 may be present; (ii) the majority of the interactions with Nsp3, i.e., the Ubl1 domain, the SUD domain and macrodomain are with RNA or ADP-ribose, and (iii) Nsp3 has been suggested to function as a scaffold mediating binding to other Nsps to form replicative

organelles explaining the lack of interactions with human proteins. Indeed, several interactions of Nsp3 with other viral Nsps have been identified by Y2H, but the functional implications of these interactions to the viral life cycle are unclear [22, 37, 38].

## Identification and characterization of inhibitors to PLpro

We employed the previously developed MALDI-TOF DUB assay [39] to evaluate the ability of 1971 FDA approved drugs to inhibit PLpro from SARS-CoV-2. Importantly the bioactivity and safety of these compounds has been tested in clinical trials allowing rapid repurposing of any hit compounds into therapies for the treatment of COVID-19. Compared to other fluorescent-based approaches, the MALDI-TOF-based assay has the advantage of using K48 ubiquitin trimer, a physiological PLpro substrate, for high-throughput screening (HTS) [40]. Further, the final products generated upon cleavage of triubiquitin are diubiquitin and monoubiquitin, which makes it compatible for analysis using the MALDI DUB assay. The reaction is terminated by the addition of 2% TFA and $^{15}$N ubiquitin. To circumvent the spot-to-spot variability typical for MALDI-TOF based detection, the mono-ubiquitin signal is normalized to the $^{15}$N internal standard signal (**Fig 3A**). The FDA-approved drug library was screened in duplicate at a concentration of 10 µM. The Z' value is commonly employed to determine the robustness of HTS, with Z' values greater than 0.5 indicating a robust assay [41]. The MALDI-TOF DUB assay resulted in an average Z' value of 0.77 ± 0.12 (**Fig 3B**). Compounds returning percentage inhibition values >50% were identified as hits and 30 compounds met this criterion (1.5% hit rate) (**Fig 3B**). The hit compounds were cherry picked and retested in a 10-point dose–response to determine $IC_{50}$ values against both PLpro and Nsp3. We used GRL-0617, a non-covalent inhibitor of both PLpro$^{CoV-1}$ [42], and PLpro$^{CoV-2}$ [9, 20, 43] as a control. GRL-0617 had been previously shown to inhibit PLpro$^{CoV-1}$ with an $IC_{50}$ of 0.6 µM [42]. We found that GRL-0617 inhibits SARS-CoV-2 PLpro with an $IC_{50}$ of 2.21 µM, which concurs nicely with the $IC_{50}$ of ~2 µM determined by recent studies [20, 43]. GRL-0617 inhibits Nsp3$^{core}$ with an $IC_{50}$ of 3.39 µM (**Fig 3F**). The MALDI-TOF-based results were further validated by orthogonal, gel-based assays employing K48-linked triUb and pro-ISG15 as substrate (**Fig 3D and 3E**; **S2A– S2C Fig**).

Based on these analyses, five compounds were confirmed for their ability to inhibit PLpro and Nsp3 *in vitro* (confirmation rate = 16.6%): Thioguanine, Nordihydroguaiaretic acid, Disulfiram, Auranofin and Tideglusib (**Fig 3C**). Thioguanine inhibits PLpro and Nsp3 with $IC_{50}$ of 4.54 and 8.67 µM respectively, in the MALDI-TOF-based assay. Intriguingly, despite this reasonable $IC_{50}$, we could not observe inhibition of PLpro by Thioguanine in the orthogonal gel-based assay Interestingly, thioguanine was previously found to inhibit PLpro$^{SARS-CoV-1}$ [44] and SARS-CoV-2 replication [45].

In our assays, we identify nordihydroguaiaretic acid (NDA) to be a potent inhibitor of PLpro and Nsp3, with $IC_{50}$ of 1.06 and 1.62 µM in the MALDI-TOF based assay. NDA is a compound extracted from plants, used in traditional medicine for the treatment of various diseases, and found to inhibit numerous cellular targets [46]. Its ability to inhibit a range of cysteine-based enzymes is linked to the redox properties of its two catechol rings, resulting in the formation of cysteine adducts [46]. Despite its beneficial effects, the main drawback to the use of this compound is its potential renal and liver toxicity over prolonged use.

Disulfiram is a thiocarbamate drug (commercially available as Antabuse®) that has received FDA approval for the treatment of alcoholism and has been in use for more than 60 years [47]. Disulfiram has been previously identified by *in silico* screening and fluorescence resonance energy transfer assay to inhibit the main SARS-CoV-2 Cys-protease, Mpro [48, 49]. Disulfiram also inhibits PLpro$^{CoV2}$ and Nsp3 with similar potencies, 0.69 and 0.66 µM in the

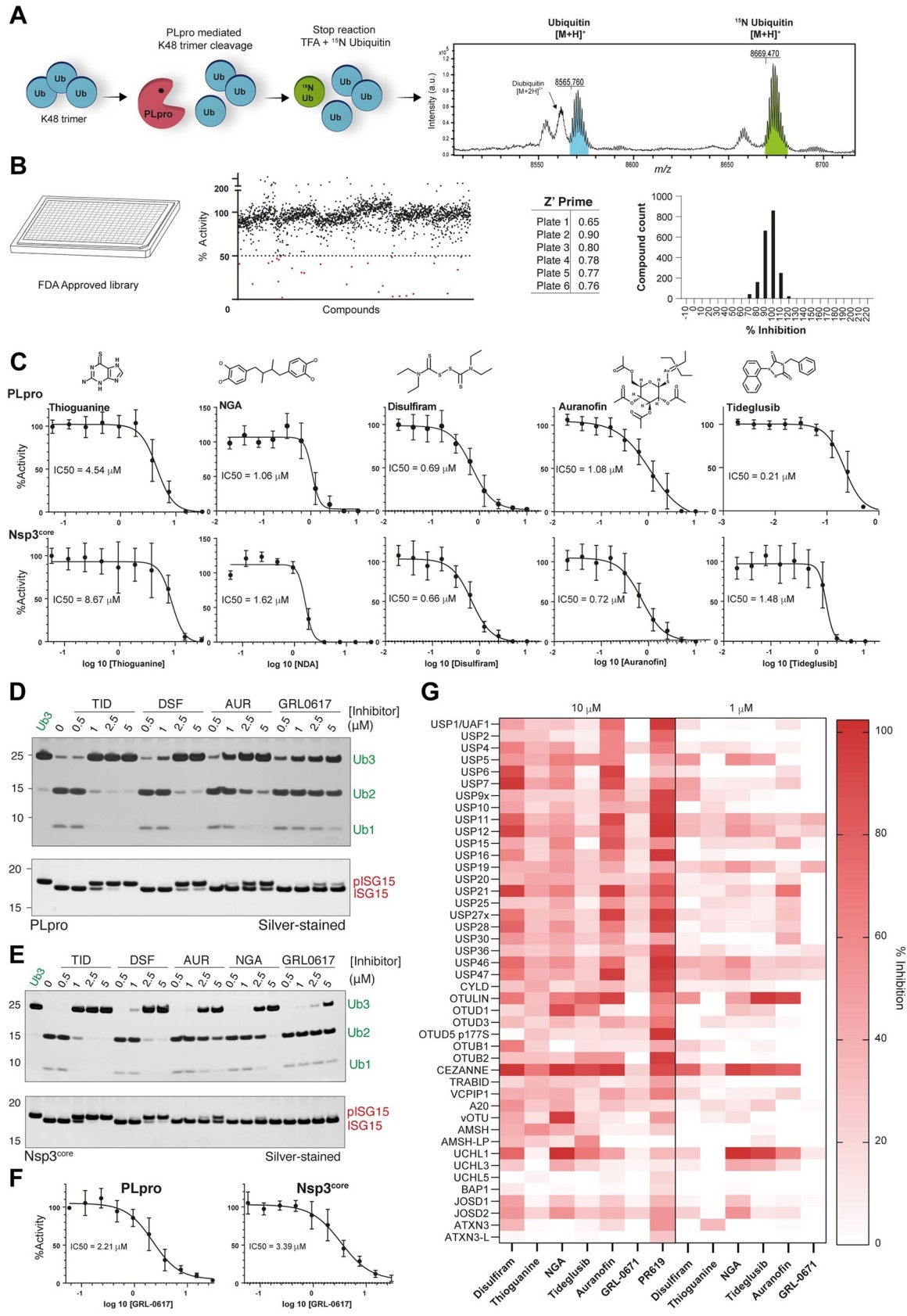

**Fig 3. Identification and characterization of PLpro inhibitors. A)** Schematic workflow of the MALDI-TOF DUB assay. K48 trimer is incubated with PLpro for 30 minutes at room temperature. The reaction is stopped by adding 2% TFA and $^{15}$N ubiquitin (as internal standard). Enzymatic activity and inhibition were assessed by MALDI-TOF MS detection of ubiquitin and $^{15}$N ubiquitin signals. **B)** FDA approved compound library screening by MALDI-TOF, Z'Prime score and HTS data distribution. **C)** IC50 calculation of compounds returning >50% inhibition values in the HTS: thioguanine, nordihydroguaiaretic acid, Disulfiram, Auranofin and Tideglusib. IC50s were calculated both against PLpro and Nsp3$^{core}$ using the MALDI-TOF DUB Assay. **D-E)** Orthogonal, gel-based assays testing PLpro and Nsp3 protease activities against K48 trimer and pro-ISG15 as substrates. Nordihydroguaiaretic acid (NDA), Disulfiram (DSF), Auranofin (AUR) and Tideglusib (TID) were tested at the indicated concentrations against PLpro and Nsp3$^{core}$. Data shown are representative of two independent experiments. **F)** IC50 calculation of GRL0617 against PLpro and Nsp3$^{core}$. **G)** Selectivity assessment of GRL-0617 and HTS positive hits against a panel of 42 human DUBs and the viral ovarian tumor (vOTU). Compounds were tested at the final concentrations of 10 and 1 μM.

MALDI-TOF-based assay, and between 1 and 2.5 μM when assessed by gel-based orthogonal assay (**Fig 3D and 3E**). Tideglusib, a Glycogen Synthase Kinase 3 (GSK3) inhibitor, has also been found to inhibit the main SARS-CoV-2 Cys-protease Mpro [48]. We found that Tideglusib strongly inhibits both PLpro and Nsp3 with IC50 of 0.21 and 1.48 μM, respectively in the MALDI-TOF based assay. Similar potency of Tidegusib was also validated in the gel-based assay (**Fig 3D and 3E**). Auranofin is a gold-containing compound employed for the treatment of rheumatoid arthritis [50], known to suppress inflammation and stimulating cell-mediated immunity. Of note, Auranofin has been recently shown to inhibit replication of SARS-CoV-2 in human cells [51]. Interestingly, we found that Auranofin strongly inhibits PLpro and Nsp3 with potency in the 1 μM range in the MALDI-TOF based assay with a similar inhibition pattern observed in the gel-based assay (**Fig 3D and 3E**).

The human genome encodes about 100 deubiquitinating enzymes, most of which are cysteine proteases [52]. Since PLpro has architectural features in common with the USP DUBs, we assessed the selectivity of GRL-0617 and the five HTS positive hits by testing them at the final concentrations of 10 and 1 μM against a DUBs panel including 42 human DUBs and the viral ovarian tumor (vOTU) from Crimean Congo Haemorrhagic Virus. Unsurprisingly, we found that Disulfiram, Thioguanine, NDA, Tideglusib and Auranofin inhibited–at different extents–several other human DUBs (**Fig 3G**), while GRL-0617 showed greater specificity with less off-target inhibition in the panel of DUBs tested (**Fig 3G**). Interestingly, GRL-0617 was also able to inhibit cleavage of the viral polypeptide (**S2D Fig**) The presence of highly reactive chemical groups able to covalently modify reactive cysteines might explain the poor specificity of these small molecules. Despite the broad range of activity of these compounds, their relative safety has been extensively evaluated in FDA pre-clinical and clinical trials.

We next tested the efficacy of these compounds to inhibit PLpro in cell-based assays measuring activation of interferon expression using an IFN-β luciferase reporter or the ability to inhibit replication of SARS-CoV-2 in CaCo cells [9]. However, none of the compounds other than GRL-0617 showed any effects of inhibiting PLpro in cells (data not shown). These results contrast with what was shown for some of these inhibitors. For instance, the use of Auranofin and thioguanine has been shown to limit SARS-CoV-2 replication [45, 51] but the underlying mechanism of action is not entirely understood. Our data demonstrate that Auranofin and thioguanine inhibit the deubiquitinating activity of the PLpro protease *in vitro*. The low potency of the identified hits at limiting viral replication suggests that repurposing may not be an effective strategy to identify inhibitors suitable for clinical development and a systematic approach is required to develop specific and effective inhibitors of PLpro/Nsp3. Our results, showing that extended versions of Nsp3 can cleave viral polypeptides, which the minimal PLpro is not as efficient at suggest that using Nsp3 to screen for inhibitors could lead to the identification of allosteric inhibitors that may inhibit all three activities of Nsp3. Despite not being very potent on their own, combining drugs targeting essential SARS-CoV-2 proteases

(PLpro and/or main protease) with other essential viral enzymes such as RNA-dependent polymerase may represent a therapeutic strategy.

## Development of nanobodies that inhibit PLpro

Since all the identified inhibitors identified show off-target effects and inhibit other human DUBs, we wanted to devise other ways to selectively inhibit PLpro as a strategy to investigate the roles of Nsp3 in the viral life cycle, in interfering with host immune signalling and to examine if inhibition of PLpro can effectively block viral replication. Previously, ubiquitin variants (Ubv) have been developed to selectively inhibit DUBs by binding to the S1 pocket [53, 54]. Hence, we aimed to engineer a nanobody (a single-domain antibody) that binds to the S1 site with high affinity to inhibit PLpro. The S1 site is important for both deubiquitinating and deISGylating activities of PLpro [17]. We therefore used a recently developed yeast surface nanobody display library to screen and identify nanobodies that specifically bind to the S1 site of PLpro [55]. To steer nanobody selection towards the S1 site, we had to discard nanobodies that recognized other regions of PLpro. Hence, we introduced a negative selection step using PLpro$^{CoV-2}$ carrying mutations in the S1 site. As these mutations were designed based on the available structure of PLpro$^{SARS}$ in complex with ubiquitin (PDB 4MM3, [8]) (**S3A Fig**), we first tested if these mutations also affected PLpro$^{CoV-2}$. Indeed, compared to WT, the mutant PLpro$^{CoV-2}$ was unreactive to propargylated Ub and was unable to cleave K48-linked triUb (**S3B and S3C Fig**). Following multiple rounds of negative and positive selection using magnetic-activated cell sorting (MACS), as outlined in **Fig 4A**, single yeast clones were tested for their binding to PLpro (**Fig 4B**). We identified several nanobodies that bound to WT PLpro-$^{CoV-2}$ but not the S1 site mutant, suggesting that these nanobodies specifically bind to the S1 site (NbSL-17, -18, 19). Sequencing these nanobodies revealed differences in the CDR1, CDR2 and CDR3 loops of the different nanobodies.

To further characterize these nanobodies, they were recombinantly expressed and purified from bacteria. Of the three individual nanobody sequences we identified, soluble expression was obtained for only two nanobodies (NbSL17 and NbSL18). We first performed analytical gel filtration, which identified NbSL18 to form a complex with PLpro$^{CoV-2}$ (**Fig 4C**). Using isothermal titration calorimetry (ITC), we find the binding affinity of NbSL18 for PLpro$^{CoV-2}$ to be ~500 nM (**Fig 4D**). We next evaluated the ability of NbSL18 to inhibit PLpro. With increasing concentration of NbSL18, the cleavage of K48-linked triUb by PLpro is inhibited with ~50% inhibition at ~500nM-1000nM NbSL18, which is consistent with the binding affinity of NbSL18 for PLpro (**Fig 4E**). In contrast, higher concentrations of NbSL18 are required to inhibit the cleavage of ISG15. Recent crystal structures of ISG15 in complex with PLpro$^{CoV-2}$ show that the C-terminal UBL of ISG15 uses a slightly different mode of interaction compared to ubiquitin [7, 9], potentially explaining the differences in the inhibition of deubiquitylating and deISGylating activities by NbSL18.

Understanding how the viral polypeptide is recognised by Nsp3 and building on this to inhibit release of mature Nsp1/2/3, could represent a legitimate strategy to suppress viral replication. Nsp1, for instance, plays a key role in the infection process by binding to host 40S ribosomal subunits and blocking the mRNA channel with its C-terminal domain, thereby suppressing translation of host proteins and sequestering ribosomes for translation of viral proteins. Preventing Nsp1 maturation from the viral polypeptide could therefore be vital in hindering viral replication [56, 57]. We therefore tested if NbSL18 could also inhibit cleavage of Nsp1-Nsp2 FL fusions. Our data reveals that NbSL18 can indeed inhibit Nsp1-2 FL cleavage by Nsp3 suggesting that the viral peptide also occupies the same S1 site as ubiquitin (**Fig 4H**).

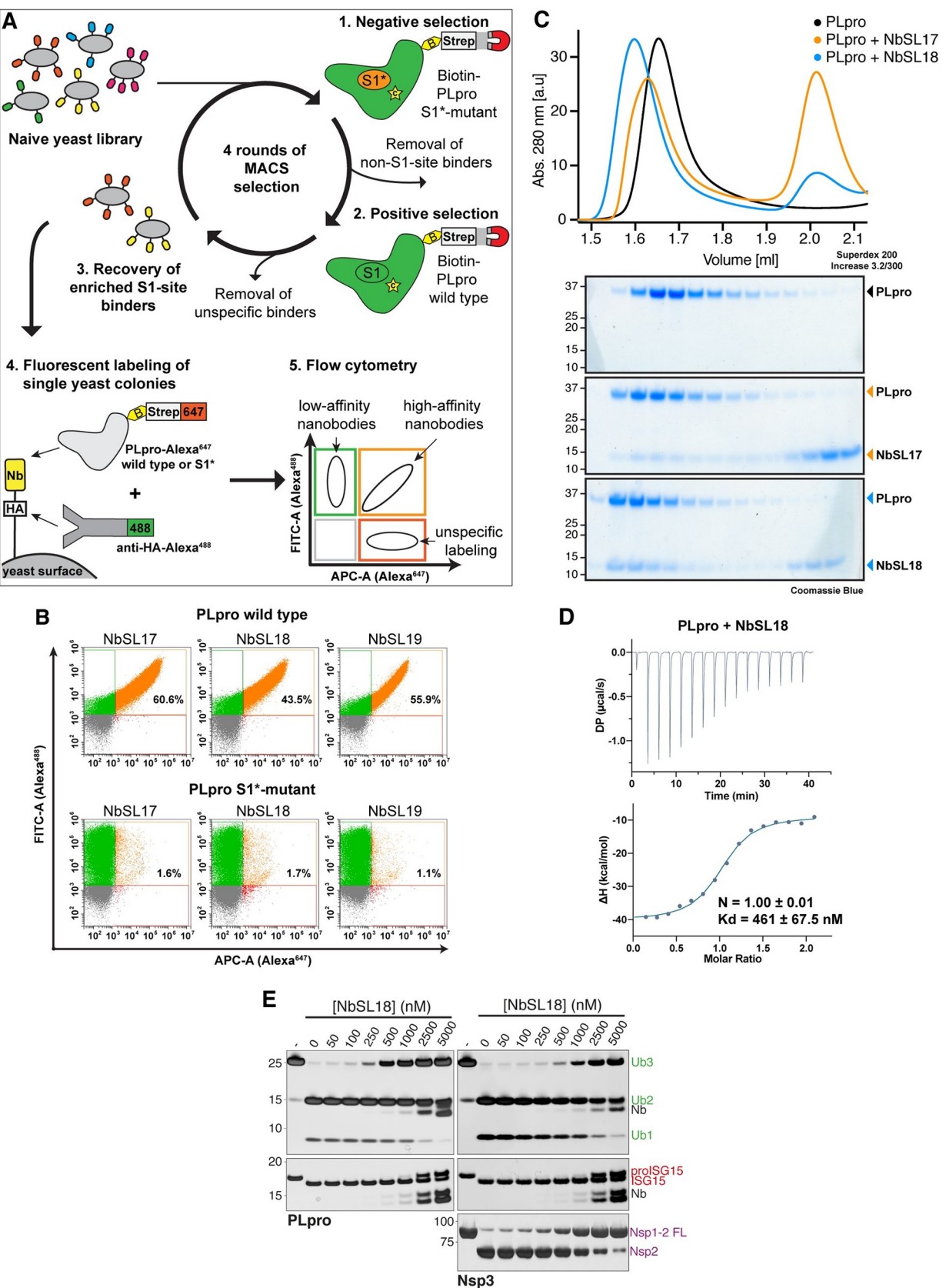

**Fig 4. Development of nanobodies that inhibit Nsp3$^{core}$. A)** Schematic workflow of yeast-surface display selection to identify site-specific nanobodies as competitive Nsp3$^{core}$ inhibitors. **B)** Flow cytometry analysis of yeast displaying anti-Nsp3 nanobodies NbSL17, NbSL18 or NbSL19, respectively. Yeast were stained with anti-HA-Alexa488 against HA-tagged Nb and Streptavidin-Alexa647 against biotinylated antigens. Top: Yeast incubated with PLpro wild-type, Bottom: Yeast incubated with PLpro S1*-mutant. Percentages indicate fraction of yeast positive for both fluorescent labels. **C)** UV-traces and coomassie-stained SDS-PAGE of analytical gel filtration analyses of PLpro-Nb complexes. PLpro alone (black), PLpro+NbSL17 (orange) and PLpro+NbSL18 (blue). **D)** Isothermal calorimetry titration measuring binding of PLpro to NbSL18. **E)** Silver stained gels of DUB assays of PLpro (left) and Nsp3$^{core}$ (right) with K48-Ub3 (top) and proISG15 (middle) and Nsp1-2 FL (bottom) as substrates in the presence of increasing concentrations of NbSL18. Silver-stained SDS-PAGE gels shown are representative of two independent experiments. Also see **S3 Fig**.

Determining a crystal structure of NbSL18 in complex with Nsp3 or PLpro would shed light into the mode of interaction and allow us to engineer mutations to not only increase the binding affinity but also selectively inhibit deubiquitylating, deISGylating or viral polypeptide processing activities. Such nanobodies will be useful to dissect the relative contributions of the different activities of PLpro [58]. However, despite several attempts we were unable to obtain well-diffracting crystals. Nevertheless, the development of these highly specific nanobodies that target PLpro represent valuable tools that could be expressed in SARS-CoV-2 infected cells to directly inhibit PLpro and thereby dissect the contributions of PLpro to viral replication and infection as shown using ubiquitin variants [59]. Further the nanobodies can be used as sensors to monitor subcellular localization of Nsp3 at different stages of viral infection. Should the nanobodies exert significant antiviral effects, they could be used as protein-based inhibitors to protect against SARS-CoV-2 and other related viruses. In summary, our work builds on extensive work characterising the PLpro domain and reveals differences in protease activities between the minimal PLpro domain of Nsp3 and longer versions of Nsp3 and highlights the need to better understand the mechanism of Nsp3. Further, recently published studies suggest that present efforts to develop inhibitors are focussed solely on PLpro [60]. Our findings advocate that it would be prudent to develop inhibitors towards Nsp3 and not just PLpro and minimally to evaluate the efficacy of compounds identified for PLpro on the ability of full length Nsp3 to cleave viral polypeptides.

## Supplemental information

Supplemental information includes S1–S3 Figs and S1 and S2 Tables.

## Materials & methods

### cDNA constructs

Human ISG15 cDNA was amplified by RT-PCR from universal human reference RNA (Agilent 740000–41), adding BamHI and NotI restriction sites at the 5' and 3' ends respectively. The digested PCR product was cloned into pGEX6P1 cut with the same enzymes to make pGEX6P1 ISG15 (DU8371).

Ubiquitin cDNA was amplified from existing clone DU8724 (pET15b 6His 3C Ubiquitin), adding restriction sites NdeI and BamHI at the 5' and 3' ends, respectively. The digested PCR product was cloned into pET24a cut with the same enzymes to make pET24a Ubiquitin (DU20027).

The cDNAs coding for SARS CoV-2 (MN908947.3) Nsp3 residues 743–1072 (PL-pro) and SARS CoV-2 Nsp3 residues 743–1072 (PL-pro) E912R E948R M953E Y1013K were synthesised by NBS Biologicals (www.nbsbio.co.uk) with 5' BamHI and 3' NotI sites. Sequences were codon optimised for bacterial expression. These were subcloned into receptor vector pETDuet 6His TEV GST 3C AVi (DU68478) digested with BamHI and NotI to make pETDuet 6His

TEV GST 3C AVi SARS CoV-2 Nsp3 PL-pro (DU67765) and pETDuet 6His TEV GST 3C AVi SARS CoV-2 Nsp3 PL-pro E912R E948R M953E Y1013K (DU67766).

pETDUET 6HIS TEV GST 3C SARS-CoV-2 Nsp3 743–1072 (PL-pro) (DU67811) was made by removing the avi tag sequence from DU67765 by PCR mutagenesis.

The cDNA coding for SARS CoV-2 Nsp1 1–180 Nsp2 1–19 8His was synthesised by GeneArt (Thermo Fisher Scientific) adding NcoI and NotI sites at the 5' and 3' ends, respectively. This sequence was subcloned into pET28a to make pET28a SARS-CoV-2 Nsp1 1–180 Nsp2 1–19 8His (DU67799).

pTXB1 Halo 3C SARS-CoV-2 (2019-nCoV) Nsp1 1-179-Intein CBD (DU67780 was made using NEBuilder (New England Biolabs), amplifying the vector from existing clone DU28033 (pTXB1-HALO-Mxe Intein-CBD) and the Nsp1 1–179 insert from existing clone DU66413 (pGEX6P1 2019-nCoV Nsp1).

pET15D Twinstrep 3C Nsp3 N179-N1329 6His (DU67831) was made using NEBuilder, amplifying the vector from existing clone DU67810 (pET15D Twinstrep 3C 6His) and the Nsp3 179–1329 insert sequence from Addgene plasmid 141257 pDONR207 SARS-CoV-2 Nsp3.

pET15D Twinstrep 3C Nsp3 1–109 Linker 179–1329 6His (DU67844) was made using NEBuilder, amplifying the vector from DU67831 and the Nsp3 1–109 with linker sequence from Addgene plasmid 141257 pDONR207 SARS-CoV-2 Nsp3.

pET15D Twinstrep 3C Nsp3 N179-N1329 Linker 1584–1944 6His (DU67847) was made using NEBuilder, amplifying the vector from pET15D Twinstrep 3C Nsp3 179–1329 6His (DU67831) and the Linker + Nsp3 1584–1944 sequence from Addgene plasmid 141257 pDONR207 SARS-CoV-2 Nsp3.

pET15D Twinstrep 3C Nsp3 A1-N1329 Linker 1584–1944 6His (DU67881) was made using NEBuilder, amplifying the vector from existing clone pET15D Twinstrep 3C-6His (DU67810) and the Nsp3 A1-N1329 Linker 1584–1944 insert from existing clone pcDNA5D FRT/TO Twinstrep 3C Nsp3 A1-N1329 Linker 1584–1944 6His IRES HA Nsp1 M1-G180 (DU67868).

pET15D Twinstrep 3C SARS-CoV-2 Nsp3 N179-N1329 6His C856A (DU67898) was made by adding the C856A mutation to clone DU67831 using PCR mutagenesis.

pET15D 8His GST 3C Nsp1-Nsp2 (DU70079) was made by reamplifying the Nsp1-Nsp2 sequence from existing clone pcDNA5 FRT/TO Sars CoV-2 Nsp1-Nsp2-Nsp3 A1-G1944-GFP (DU70082) (generated by NEBuilder), directly digesting with BamHI and NotI and cloning into existing plasmid pET15D 8His GST (DU70054) cut with the same enzymes.

pcDNA5D FRT/TO mCherry (DU41458) was made by amplifying the mCherry sequence plus mcs and cloning into the BspTI and NotI sites of pcDNA5D FRT/TO (DU41459), a derivative of pcDNA5 FRT/TO (Thermo Fisher).

pcDNA5D FRT/TO Twinstrep 3C Nsp3 A1-N1329 Linker 1584–1944 6His (DU67879) was made by reamplifying Twinstrep 3C Nsp3 A1-N1329 Linker 1584–1944 6His from pcDNA5D FRT TO Twinstrep 3C Nsp3 A1-N1329 Linker 1584–1944 6His IRES HA Nsp1 M1-G180 (DU67868) (made by NEBuilder) and cloning into pcDNA5D FRT/TO (DU41459) BglII/NotI into BamHI/NotI.

pcDNA5D FRT/TO Twinstrep 3C SARS CoV-2 Nsp3 743–1072 (PL-pro) (DU67916) was made by removing the HA and replacing with Twinstrep 3C using Q5 site-directed mutagenesis (New England Biolabs E0552S) in pcDNA5D FRT/TO HA SARS-CoV-2 Nsp3 743–1072 (PL-pro) (DU67767). pcDNA5D FRT/TO HA SARS-CoV-2 Nsp3 743–1072 (PL-pro) (DU67767) was made by cloning the synthesised SARS-CoV-2 Nsp3 743–1072 (PL-pro) sequence into pcDNA5D FRT/TO HA (DU41456) using BamHI/NotI.

All PCR reactions were carried out using KOD Hot Start DNA polymerase (Novagen).

All RT-PCRs were carried out using Takara's PrimeScript High Fidelity RT-PCR kit (R022A).

DNA sequencing was performed by MRC PPU DNA Sequencing and Services, School of Life Sciences, University of Dundee (www.dnaseq.co.uk).

See **S1 Table** for summary of constructs used in this study. All constructs described here can be obtained from https://mrcppu-covid.bio/

## Protein expression and purification

All recombinant proteins were expressed in *E. coli* BL21-CodonPlus (DE3) cells. Bacterial cultures were grown in 2x TY media supplemented with 100 μg/ml ampicillin to an $OD_{600}$ of 0.6–0.8 at 37˚C in shaking incubators. Protein expression was induced by the addition of 300 μM IPTG followed by overnight incubation at 18˚C. Cells were harvested by centrifugation at 3500 g for 30 mins and pellets were suspended in lysis buffer (50 mM Tris-HCl pH 7.5, 300 mM NaCl, 10% glycerol, 0.075% v/v BME, 1 mM benzamidine, 1 mM AEBSF and a complete protease inhibitor cocktail tablet (Roche). Resuspended cells were then lysed by sonication and clarified by centrifugation at 35,000g for 45 min at 4˚C.

**His$_6$-GST-PLpro.** Clarified lysate was incubated with His60 Ni Superflow Resin (TaKaRa) for 1 hr at 4˚C. Resin was washed with buffer (300 mM NaCl, 50 mM Tris pH 7.5) supplemented with increasing concentrations of imidazole. Protein was eluted with 300 mM Imidazole. The eluent was then incubated with GSH Sepharose (TaKaRa) for 1 hour before being washed by high salt buffer (500 mM NaCl, 50 mM Tris pH 7.5, 5 10 mM DTT) and then low salt buffer (150 mM NaCl, 50 mM Tris pH 7.5, 1 mM DTT). Resin was incubated with low salt buffer supplemented with PreScission protease o/n at 4˚C to cleave the GST tag. Protein was eluted and concentrated (Amicon Centrifugal Filter Units, Merck) followed by size exclusion chromatography (Superdex 75 16/60, Cytiva) in SEC buffer (20 mM Tris-HCl pH 7.5, 150 mM NaCl, 1 mM DTT).

**TwinStrep-Nsp3$^{core}$.** The clarified lysate was then filtered through a 0.45-micron syringe filter and purified by affinity chromatography (StrepTrap, Cytiva). The column was washed thoroughly with buffer S (50 mM Tris-HCl pH 7.5, 300 mM NaCl, 1 mM DTT) and protein was eluted with buffer S supplemented with 10 mM desthiobiotin. The protein was then further purified by anion exchange chromatography (Resource Q, Cytiva) and eluted in a gradient with buffer Q (50 mM Tris-HCl pH 8.5, 1 M NaCl, 1 mM DTT), followed by size exclusion chromatography (Superdex 200 16/60, Cytiva) in SEC buffer (20 mM Tris-HCl pH 7.5, 300 mM NaCl, 1 mM DTT).

**K48-Ub3.** Untagged monoubiquitin was purified as described previously and polyubiquitin chains were synthesized and purified as described previously [61].

**proISG15.** Clarified lysate was incubated with GSH sepharose for 1hr. Resin was then washed with high salt buffer (500 mM NaCl, 50 mM Tris pH 7.5, 5 10 mM DTT) and then low salt buffer (150 mM NaCl, 50 mM Tris pH 7.5, 1 mM DTT). Resin was incubated with low salt buffer supplemented with PreScission protease o/n at 4˚C to cleave the GST tag. Protein was eluted and concentrated (Amicon Centrifugal Filter Units, Merck) followed by size exclusion chromatography (Superdex 75 16/60, Cytiva) in 20 mM Tris-HCl pH 7.5, 150 mM NaCl, 1 mM DTT.

**Nsp1-2Δ-His$_8$.** Clarified lysate was incubated with His60 Ni Superflow Resin (TaKaRa). Resin was washed with a buffer (300 mM NaCl, 50 mM Tris pH 7.5) supplemented with increasing concentrations of imidazole. Protein was eluted with 300 mM Imidazole. Protein was eluted and concentrated (Amicon Centrifugal Filter Units, Merck) followed by size

exclusion chromatography (Superdex 75 16/60, Cytiva) in a SEC buffer (20 mM Tris-HCl pH 7.5, 150 mM NaCl, 1 mM DTT).

**Nsp1-2 FL.** Clarified lysate was incubated with His60 Ni Superflow Resin (TaKaRa) for 1 hr at 4˚C. Resin was washed with buffer (300 mM NaCl, 50 mM Tris pH 7.5) supplemented with increasing concentrations of imidazole. Protein was eluted with 300 mM Imidazole. The eluent was then incubated with GSH Sepharose (TaKaRa) for 1 hour before being washed by high salt buffer (500 mM NaCl, 50 mM Tris pH 7.5, 5 10 mM DTT) and then low salt buffer (150 mM NaCl, 50 mM Tris pH 7.5, 1 mM DTT). Resin was incubated with low salt buffer supplemented with PreScission protease o/n at 4˚C to cleave the GST tag. Protein was collected, loaded onto a 6 mL Resource Q column, and eluted in a gradient of buffer Q (50 mM Tris pH 8.5, 1000 mM NaCl). Peak fractions were concentrated (Amicon Centrifugal Filter Units, Merck) followed by size exclusion chromatography (Superdex 75 16/60, Cytiva) in a SEC buffer (20 mM Tris-HCl pH 7.5, 150 mM NaCl, 1 mM DTT).

## Qualitative protease assays

PLpro and Nsp3$^{core}$ were diluted in buffer A (50 mM Tris-HCl pH 7.5, 50 mM NaCl, 10 mM DTT) and incubated at RT for 10 minutes to fully reduce the catalytic cysteine. For most protease cleavage assays, 3 µM substrate was incubated with varying concentrations of PLpro and Nsp3$^{core}$. For assays comparing the relative activities of each protease against a panel of substrates and assessing the linkage specificity, 50 nM of Plpro/Nsp3 was used. For assays comparing the relative activities of PLpro and Nsp3 with Nsp1-2 FL, Ub3, proISG15 and Nsp1-2Δ the respective enzyme concentrations were 50 nM (Nsp1-2 FL, Ub3), 20 nM (proISG15) and 1 µM (Nsp1-2Δ). For the assay comparing the cleavage of Nsp1-2(Δ) G180A relative to WT by Nsp3$^{core}$ and the assay comparing cleavage of Nsp1-2Δ by all Nsp3 constructs, 5 µM protease was incubated with 10 µM substrate. For the assay in which increasing concentrations of PLpro were incubated with Nsp1-2Δ, 10 µM substrate was used. All assays were carried out at 30˚C and quenched at indicated time points by adding LDS loading dye. The samples were separated on 4–12% SDS-PAGE gel (Life Technology) and silver stained using Pierce Silver stain kit (Thermo Fisher).

## Preparation of Nsp3-Nsp1$^{prg}$ complex

HALO-Nsp1 thioester was generated following intein cleavage as described previously for ubiquitin [62]. The HALO-Nsp1-thioester was buffer exchanged into HEPES buffer (pH 8.0) and reacted with 250 mM propargylamine (Sigma) for 3 hours at RT. The reactive probe was then separated from the excess propargylamine by size-exclusion chromatography (Superdex 200 16/60, Cytiva) in 1x PBS. To test reactivity of the probe (S1B), 5 µM Nsp3$^{core}$/PLpro was incubated with 10 µM of HALO-Nsp1$^{prg}$ at 30˚C for 1 hr. The reaction was quenched with 4x LDS dye, resolved on a 4–12% SDS-PAGE gel (Life Technology) and stained with InstantBlue (Expedeon).

## Small molecule/nanobody inhibition assays

PLpro and Nsp3$^{core}$ were diluted in buffer I (50 mM Tris-HCl pH 7.5, 50 mM NaCl, 0.01% (w/v) BSA). The DUB was pre-incubated with the indicated concentration of small molecule inhibitor/nanobody for 30 min at RT. A final concentration of 50 nM DUB was incubated with 3 µM substrate at 30˚C for 1 hour. The assay was quenched at indicated time points by adding LDS loading dye. The samples were separated on 4–12% SDS-PAGE gel (Life Technology) and silver stained using Pierce Silver stain kit (Thermo Fisher).

## Nanobody selection

**Purification and biotinylation of PLpro protein for nanobody selection.** Clarified lysates containing 6His-GST-AVI-PLpro or 6His-GST-AVI-PLpro E912R E948R M953E Y1013K (S1* mutant) were incubated with His60 Ni Superflow Resin (TaKaRa) for 1 hr at 4˚C. Resin was washed with 300 mM NaCl, 50 mM Tris pH 7.5, 10 mM Imidazole, 2 mM BME and protein was eluted with added 300 mM Imidazole. The eluted protein was concentrated to ~100 μM and biotinylated by incubation with 2 μM BirA, 5 mM $MgCl_2$, 2 mM ATP and 250 μM Biotin for 1 h at 25˚C. Additional 250 μM Biotin were added and the reaction was incubated for 1 h at 25˚C. The reaction mix was then incubated with GSH Sepharose beads for 1 hour at 4˚C and the resin was washed with 50 mM Tris pH 7.5, 300 mM NaCl, 2 mM BME. The 6His-GST-tag was removed by incubation with PreScission protease overnight at 4˚C and the biotinylated protein was eluted from the resin, concentrated to ~1 mg/ml, flash-frozen in liquid nitrogen and stored at –80˚C. Successful biotinylation of the target protein was assessed through incubation with 5-fold excess streptavidin for 5 min at 20˚C, followed by SDS-PAGE analysis of the induced MW-shift of the streptavidin complex.

**Selection of synthetic nanobodies by yeast surface display.** Site-specific nanobodies against PLpro were selected from the naïve yeast display nanobody library, which was developed and generously provided by the Kruse lab [55] Yeast were grown in YGLC-glu medium (80 mM sodium citrate pH 4.5 (Sigma), 6.7 g/l yeast nitrogen base w/o amino acids (BD Biosciences), 2% glucose, 3.8 g/l Do mix-trp at 30˚C and 200 rpm shaking overnight and nanobody expression was induced by incubation in YGLC-gal medium (recipe as YGLC-glu but with galactose in place of glucose) at 20˚C and 200 rpm for 48–72 h. Nanobody-expressing yeast were washed with and selected against in sterile-filtered, ice-cold PBM (1x PBS (Gibco) supplemented with 0.5% BSA (Sigma) and 5 mM Maltose (Sigma)). Nanobodies were selected in four rounds of magnetic cell sorting (MACS) on LD columns (Miltenyi) for negative selection against biotinylated PLpro S1* mutant and subsequent positive selection on LS columns (Miltenyi) against biotinylated PLpro. In the first MACS round, 5*10^9 yeast in 5 ml PBE were incubated with 400 nM biotinylated PLpro S1* mutant immobilized on Streptavidin microbeads (Miltenyi) for 40 min at 4˚C. The mixture was applied to an equilibrated LD column on a magnetic rack, the flow-through was collected and the column was washed with 2 ml PBM to collect further unbound yeast. Unbound yeast was pelleted at 4˚C and 2500 g for 3 min and resuspended in 5 ml PBM containing 400 nM biotinylated PLpro immobilised on streptavidin microbeads and rotated for 1 h at 4˚C. Yeast were pelleted as above, resuspended in 3 ml PBM and applied to an LS column on a magnetic rack. The column was washed three times with 3 ml PBM, removed from the magnetic rack and yeast were eluted with 5 ml PBM. Eluted yeast were pelleted, resuspended in 4 ml YGLC-glu and incubated at 30˚C and 200 rpm shaking overnight. The next day, yeast were used to inoculate 4 ml YGLC-gal at OD600 = 1 and incubated at 20˚C and 200 rpm for 48–72 h. For the second MACS selection rounds, 1*10^8 yeast were pelleted and resuspended in 500 μl PBM containing 100 nM Streptavidin-Alexa Fluor 647 conjugate (Thermo Fisher) and 400 nM PLpro S1*-mutant and incubated for 40 min at 4˚C. Yeast were washed with 1 ml PBM and resuspended in 950 μl PBM + 50 μl anti-Alexa647 microbeads (Miltenyi) and rotated for 20 min at 4˚C. Yeast were negatively selected on LD column as described above. Flow-through yeast were washed once with 1 ml PBM and resuspended 500 μl PBM containing 100 nM Streptavidin-Alexa647 and 400 nM biotinylated PLpro and incubated for 2 h at 4˚C. Yeast were washed with 1 ml PBM, resuspended in 950 μl PBM + 50 μl anti-Alexa647 microbeads and rotated for 20 min at 4˚C. Yeast were applied to and eluted from LS column for positive selection as described above. Yeast were recovered in YGLC-glu and incubated in YGLC-gal as above. The negative selection of the third and fourth

MACS round were identical to the second round described above. For the positive selection of the third MACS round, the flow-through yeast was incubated with 2000 nM PLpro in 500 μl PBM for 1 h at 4˚C, washed twice with 1 ml PBM, resuspended in 50 nM Streptavidin-647 in 500 ml PBM and rotated for 15 min at 4˚C. Yeast was pelleted and resuspended in 5 ml PBM and positively selected on LS columns as described above. The fourth MACS selection round was identical to the third round, but with reduced PLpro concentration (500 nM) for positive selection. The selection was monitored by flow cytometry on a cytoFLEX S (Beckman) by analysis of fluorescent signals of antigen-bound (Streptavidin-647) and total nanobodies (anti-HA-Alexa488, Cell Signaling Technologies) displayed on the yeast surface.

**Sequencing and subcloning of nanobodies.** Yeast obtained after four rounds of MACS selection was plated on YGLC-glu agar to obtain single colonies after 96 h at 30˚C. Single yeast colonies were picked and resuspended in 100 μl 200 mM lithium acetate and 1% SDS each. The cell suspension was incubated for 5 min at 70˚C and subsequently mixed with 300 μl ethanol and vortexed. The mixture was centrifuged at 15k g for 3 min, the pellet was washed with 70% ethanol and resuspended in 100 μl H2O. Cell debris was removed by centrifugation at 15k g for 1 min and 1 μl of the supernatant was used as template DNA for 25 μl PCR reaction (KOD HotStart, Merck) with primers NbLib-fwd-i (CAGCTGCAGGAAAGCGGCGG) and NbLib-rev-i (GCTGCTCACGGTCACCTGG) to amplify the nanobody insert sequence. Nanobody PCR fragments were analyzed on 2% TAE-agarose gels and purified with the QIAquick gel extraction kit (Qiagen). PCR fragments were sequenced and individual nanobodies were cloned using the In-Fusion cloning kit (Takara Bio) into a pET28a vector linearised by PCR with primers NbLib_pET28a_fwd (GGTGACCGTGAGCAGCCACCACCACCACCA CCACTGA GATCCGGCTGCTAACAAAGC) and NbLib_pET28a_rev (CCGCCG CTTTCCTGCAGCTGCACCTGCGCCATCGCCGGCT).

## ITC measurement

ITC measurements were performed on a MicroCal PEAQ-ITC instrument (Malvern) at 25˚C. All proteins were dialyzed overnight at 4˚C into 50 mM Tris pH 7.5, 150 mM NaCl and 0.2 mM TCEP prior to ITC analysis. NbSL18 at 148 μM was loaded in the syringe and titrated in 16 2.5 μl injections to 14.4 μM PLpro in the cell. ITC data was fitted with a one-sided binding model (MicroCal Analysis Software, Malvern) to calculate binding constants.

## Transfections, lysis and co-immunoprecipitations

A549 human alveolar epithelial cells were transfected with empty vector in triplicate or with vectors expressing SARS CoV-2 Nsp3ΔTM or PLpro fused to a 2xStrep tag at the N-terminus (4 replicates each). 12hr later, cells were stimulated with 50 ng/ml IFN-a (Cell Signalling Technologies) for 36hr. Cells were lysed in ice-cold IP buffer (50 mM Tris-HCl, pH7.5, 150 mM NaCl, 0.5% NP-40, 1 mM AEBSF) and clarified by centrifugation. Viral proteins were affinity purified from 4 mg lysates by tumbling with 40 μl Strep-Tactin Sepharose (IBA Life Sciences) for 90 min at 4˚C. Beads were washed two times with IP buffer containing 500 mM NaCl and 0.5% NP-40 and twice more with IP buffer without detergent. 90% of the final wash was transferred to a fresh LoBind tube (Eppendorf) for processing for LC-MS/MS analysis. The remaining 10% was reserved to confirm viral protein expression and pulldown by Western blot for Nsp3. Efficient IFN-a stimulation was confirmed by blotting lysates for ISG15.

## On-bead digestion and TMT labeling

The beads from HA-Nsp3, HA-PLpro and control immunoprecipitations were resuspended in 100 μl of 100 mM Tris-Cl (pH 8.0). The proteins were reduced by adding final 5mM DTT and

incubated on a Thermomixer with a gentle agitation at 56°C for 20 minutes. Following, the samples were brought to room temperature and alkylated by adding final 20 mM Iodoaceta-mide and incubated in dark at room temperature on a Thermomixer with a gentle agitation for 30 minutes. Samples were subjected to on-bead tryptic digestion by adding a digestion buffer (2 M Urea + 500 ng of sequencing grade trypsin per sample) and incubated at 30°C tem-perature on a Thermomixer for about 16hrs. Post digestion, samples were centrifuged (At room temperature at 2000 g for 3 minutes) and the supernatant was collected into a new 1.5ml collection tubes and acidified by adding final 1% TFA. Samples were then subjected SDB-RP peptide clean up using in-house prepared SDB-RP stage-tips. Briefly, two 16guage SDB-RP disks were packed into a 250 ul of pipette tips. The stage-tips were activated by adding 150ul of loading buffer (1%TFA in Isopropanol vol/vol) and centrifuged at 2,500 g at RT for 10 min-utes. Subsequently, the acidified peptide digest was loaded and centrifuged at 2,500 g at RT for 10 minutes. The flow-through was reapplied and centrifuged. The stage-tips were washed once with loading buffer and washed again with wash buffer (3% ACN in 0.3% TFA vol/vol). Finally, samples were eluted by adding 60 ul elution buffer I (50% ACN in 1.25% NH4OH vol/vol) and eluted again by adding 60ul of elution buffer II (80% ACN in 1.25& NH4OH). The eluates were immediately snap frozen and vacuum dried in a speed vac.

**TMT labeling.** The Vacuum dried peptides are reconstituted in 17 µl of 50mM TEABC (pH 8.0). 3ul of anhydrous ACN and 4 ul of TMT reagent (32ug of TMT reporter tag) of each TMT reporter tag (TMT 10 plex kit, Product number: 90110 and A37725., Thermo Fisher Sci-entific) was added to each sample and incubated at room temperature on a Thermo mixer with a gentle agitation (600 rpm) for 2 hrs. The TMT labelling has been performed to each sample as, HA-PLpro- sample 1–3 was labelled to 126, 127N and 127C and HA-Nsp3 sample 1–4 = 3 was labelled to 128N, 129C and 129N and HA-Empty sample 1–3 was labelled 129C, 130N and 130C reporter tags. Following, another 25 µl of 50mM TEABC was added to each sample and incubated further 10 minutes. TMT labeling efficiency was verified by taking a 5 µl from each sample pooled into a new collection tube and immediately vacuum dried and desalted using C18 Stage-tip as explained previously [63]. TMT labeling efficiency was identi-fied to be >99%. Further, each sample was quenched by adding 2.5 µl of 5% Hydroxyl amine and incubated at room temperature for 10 minutes. Following, samples were pooled and vac-uum dried in a speed-vac. Pooled TMT labelled peptide digest was subjected to mini high pH reversed-phase micro-column-based fractionation as described [63, 64]. A total of 8 fractions were prepared and concatenated into final four fractions (FR1+FR4, FR2+FR5, FR3+FR7 and FR4+FR8), vacuum dried and subjected to LC-MS/MS analysis.

**LC-MS/MS analysis.** The pooled micro-column based bRPLC fractions were reconsti-tuted in 15 ul of 0.1% formic acid and 3% ACN buffer and subjected to LC-MS/MS/MS analy-sis on Orbitrap Fusion Lumos Tribrid mass spectrometer that is interfaced with 3000 RSLC nano liquid chromatography system. Sample was loaded on to a 2 cm trapping column (Pep-Map C18 100A – 300 µm. Part number: 160454. Thermo Fisher Scientific) at 5 ul/min flow rate using loading pump and analyzed on a 50cm analytical column (EASY-Spray column, 50 cm × 75 µm ID, Part number: ES803) at 300 nl/min flow rate that is interfaced to the mass spectrometer using Easy nLC source and electro sprayed directly into the mass spectrometer. LC gradient was applied from a 3% to 30% of solvent-B at 300 nl/min flow rate (Solvent-B: 80% ACN) for 105 minutes and 30% to 40% solvent-B for 15 minutes and 35% to 99% Sol-vent-B for 5 minutes which is maintained at 90% B for 10 minutes and washed the column at 3% solvent-B for another 10 minutes comprising a total of 140 min run with a 120-minute gra-dient in a data dependent MS3 mode (FT- IT CID MS2-FT MS3). The full scan MS1 was acquired at a resolution of 120,000 at m/z 200 and measured using ultra-high filed Orbitrap mass analyzer. The top 10 precursor ions were targeted for the MS2 and top 10 fragment ions

were isolated by enabling synchronous precursor selection for MS3 and analyzed using Ultra high-filed Orbitrap mass analyzer. Precursor ions were isolated using quadrupole mass filter with 0.7 Da mass window and MS2 spectra was fragmented using collisional induced dissociation and recorded using Iontrap mass analyzer in a rapid mode. Top 10 fragment ions with a 1.2 Da isolation window was isolated and fragmented using beam-type collisional induced dissociation (HCD) at 65% normalized collision energy and analyzed using Ultra-high filed Orbitrap mass analyzer using 50,000 resolution at m/z 200. The AGC targets were set as 2E5 for MS1, 2E4 for MS2 and 5E4 for MS3 with 100 ms, 54 ms and 120 ms ion injections respectively.

## Mass spectrometry data analysis and Bioinformatics analysis

The MaxQuant software suite [65] version 1.6.10.0 was used for database search with the following parameter. Reporter ion MS3 type: 9plex TMT at a reporter ion mass tolerance of 0.003 Da. The TMT isotopic reporter ion correction factors were manually entered as provided by the manufacturer. The following group specific parameters were used, A built-in Andromeda search engine was used by specifying Trypsin/P as a specific enzyme by allowing 2 missed cleavages, minimum length of 7 amino acids, Oxidation of (M), Acetyl (Protein-N-terminal), Deamidation N and Q were selected as variable modifications. Carbamidomethylation Cys was selected as fixed modification. First search tolerance of 20 ppm and main search tolerance of 4.5 ppm were selected. Global Parameters: Uniprot Human protein database appended with SARS-COV-2 NSP3 with twin strep-tag (release 2017–02; 42,101 sequences) was used for the database search and 1% peptide and protein level FDR was applied. For protein quantification, min ratio count was set at 2 for accurate reporter ion quantification. The MaxQuant output protein group text files were processed using Perseus software suite [65], version 1.6.10.45 was used. The data was filtered for any proteins that are flagged as common contaminants and reverse hits. The TMT reporter ion intensities were log2 transformed and subsequently the TMT reporter tags of PLpro, NSP3 and control conditions were categorized to perform statistical analysis. Two sample Welch's T-test was performed by applying 5% permutation-based FDR to identify the differentially enriched and significant protein groups between HA-PLPro vs control and HA-NSP3 vs control groups. The mass spectrometry proteomics data have been deposited to the ProteomeXchange Consortium via the PRIDE [66] partner repository with the dataset identifier PXD022904.

## MALDI TOF assays

MTP Anchor Chip 1536 BC target plates (# 8280787) and MTP target frame (# 8074115) were purchased from Bruker Daltonics (Billerica, MA). 2',5'- Dihydroxyacetophenone matrix (DHAP matrix) was purchased from Tokio Chemical Industries (Product Number: D1955). LC-MS grade solvents and reagents were purchased from different commercial vendors: Acetonitrile (Merck-Millipore—Cat. No: 1.00030.2500), Ammonium Citrate dibasic (Fluka–Cat. No: 09833), DMSO (Sigma-Aldrich—Cat. No: D8418), Trifluoroacetic Acid–TFA (Thermo Scientific—Cat. Number 85183). Assay were performed in 384 well microplate, PS, small volume, hibase, white (Greiner bio-one–Cat. Number 784904) and covered using Silverseal Sealer Aluminium Foil (Greiner Bio-One—Cat. Number 676090). To prepare DHAP matrix solution, 25 mg diammonium hydrogen citrate were dissolved in 1 ml of milliQ water. 7.6 mg of DHAP matrix were dissolved in a solution of 375 ul of LC-MS grade Ethanol and 125 uL of diammonium hydrogen citrate solution with the aid of vigorous vortexing.

Enzymatic reactions and MALDI-TOF High-Throughput Screening (HTS) were set up in assay as previously described [40]. Briefly, FDA approved compound library was delivered into 384 well plated plates using non-contact acoustic delivering. Two columns were reserved

for positive (DMSO only) and negative controls (no enzyme). PLpro and K48 ubiquitin trimer were diluted in buffer containing 40 mM Tris (pH 7.5), and 0.01% BSA at a final concentration of 1.2 ng/μl and 0.2 mg/ml, respectively. 3 μl of PLpro solution was aliquoted into 384 well plates using FluidX XRD system equipped with 16 channel tubing cartridges. Plates were centrifuged for 1 minute at 1000 x g, covered using aluminium sealing foil to prevent evaporation and incubate at room temperature for 30 minutes. Reaction was started by adding 3 μl of K48 ubiquitin trimer on the entire assay plate and incubated at room temperature for 30 minutes. The reaction was stopped by adding 3 μl TFA—final concentration of 2% (v/v) supplemented with 4 μM $^{15}$N labelled ubiquitin. A TTP Labtech Mosquito® HTS equipped with 5 positions plate deck was employed to mix sample and DHAP matrix and to spot the mixture on the MALDI target. Four 384-well plates were combined into one Anchor Chip 1536 BC MALDI-TOF target plates. Spotted and dried MALDI target was subjected to automatic MALDI-TOF MS analysis. Mass spectra were acquired with a RapifleX MALDI-TOF/TOF instrument from Bruker Daltonics equipped with Compass software for FlexSeries 2.0. Mass spectra were acquired in the mass range of m/z 8000–9200, to include ubiquitin (8565.76 kDa) and the $^{15}$N ubiquitin (8669.47 kDa) signals. 4000 laser shots per sample spot were accumulated in positive ionization mode using a 500 ns pulsed ion extraction with a 10 kHz laser frequency, a digitizer setting of 5.00 GS/s, and a single Smartbeam parameter at a 800 μm scan range. The acquired spectra were processed with a centroid peak detection set to a signal-to-noise ratio (S/N) of 5 and Gaussian smoothing (0.5 m/z; 10 cycle). The signal of $^{15}$N ubiquitin was added to the mass control list and used as internal calibrant. MALDI-TOF data, processed with flexAnalysis were exported as a comma delimited (.csv) file. Data sets were further processed using Excel or GraphPad Prism (v7.03; GraphPad Software, La Jolla, CA). PLpro activity was tracked using the measured area (signal) of the enzymatic product (ubiquitin) and the corresponding labelled internal standard ($^{15}$N ubiquitin). All data points were reported as ratio of ubiquitin signal over the internal standard signal. Generated sample data were expressed as percentage of control (PoC) according to the following equation:

$$PoC = \left(\frac{Sample - b}{c - b}\right) x\, 100$$

Were b being the average of the negative controls and c the average of the positive controls.

To assess the robustness and quality of the hit discovery campaign a key metric, the Z' parameter [41] was employed as for following equation:

$$Z - Prime = 1 - \frac{3\left(\sigma_{p + \sigma_n}\right)}{\mu_{p - \mu_n}}$$

Were $\sigma_p$, $\sigma_n$ being positive and negative control standard deviations and $\mu_p$, $\mu_n$ the difference of the averages of positive and negative controls. GraphPad Prism was utilized for graphical representation of the data.

FDA approved compound Set: Apex bio DiscoveryProbe™ FDA-approved Drug Library (Catalog No. L1021) is a unique collection of 1971 small molecule, FDA approved drugs with known bioavailability and safety data in humans.

## Supporting information

**S1 Fig. Characterisation of Nsp3. A)** Assay showing the cleavage of Nsp1-2Δ by the indicated PLpro (1) and Nsp3 constructs (2–5) (colour-coding as in Fig 1A). **B)** DUB assay showing the polyubiquitin linkage specificity of PLpro. **C)** Assay showing the effect of increasing PLpro

concentration on cleavage of Nsp1-2Δ visualized by Coomassie staining. **D)** Sequence alignment of the C-termini of all PLpro substrates highlighting the cleavage site and consensus recognition motif. **E)** Cleavage assay comparing activity of WT and C856A Nsp3$^{core}$ against K48-linked Ub3. **F)** Assay monitoring the conversion of PLpro and Nsp3 by a HALO-Nsp1-prg probe. **G)** Michaelis-Menten kinetics of K48-Ub3 cleavage by PLpro and Nsp3$^{core}$.
(PDF)

**S2 Fig. Characterization of inhibitors.** Orthogonal, gel-based assays employing K48 trimer (**A, B**) and pro-ISG15 (**C**). Tideglusib (TID), Disulfiram (DSF), Nordihydroguaiaretic acid (NDA), Thioguanine (6TG), Methimazole (MIZ), Bacitracin (BAC), Lenalidomide (LEN), Auranofin (AUR), and Iniparib (INI) were tested at the indicated concentrations against PLpro (**A**) and NSP3 (**B,C**). **D)** Assay testing the ability of GRL0617 to inhibit Nsp1-2 FL cleavage by Nsp3$^{core}$. Data shown are representative of two independent experiments.
(PDF)

**S3 Fig. Design of PLpro S1\*-mutant and identified inhibitory nanobody sequences. A)** Structure of SARS-CoV PLpro (blue) bound to Ubiquitin (green) (PDB 4MM3). Stick models show mutated S1-binding site residues (gold) and catalytic cysteine C856. Labels indicate right-hand architecture of PLpro with Fingers, Palm and Thumb subdomains. **B)** Coomassie stained gel comparing reactivity of PLpro WT and the S1\* mutant with Ub-prg. **C)** DUB assay of PLpro WT and the S1\* mutant with K48-Ub3 as substrate visualized by silver staining. **D-E)** DUB assays of PLpro (D) and Nsp3$^{core}$ (E) with K48-Ub3 (top) and proISG15 (bottom) in the presence of increasing concentrations of NbSL17. **F)** Sequence alignment of identified inhibitory nanobodies NbSL17, NbSL18 and NbSL19 with synthetic nanobody Nb.201 (PDB 5VNV). Secondary structure elements and location of complementarity determining regions (CDRs) are indicated above the sequence.
(PDF)

**S1 Table. List of cDNA constructs.**
(PDF)

**S2 Table. Details of reagents.**
(PDF)

## Acknowledgments

We thank members of the Kulathu lab for discussions and help, A. Knebel, D. Milrine and M. McFarland for discussions, analyses, and helpful comments. We acknowledge MRC Reagents and Services for providing valuable reagents and support, Jason Swedlow and the National Phenotyping Screening Centre for inhibitor screens, Ignacio Moraga for help with the nanobody selection. Anthony Hope passed away before the submission of the final version of this manuscript. YK accepts responsibility for the integrity and validity of the data collected and analyzed. This work is dedicated to the memory of Anthony Hope who was passionate about this project and identifying potent inhibitors of PLpro to treat COVID-19.

## Author Contributions

**Conceptualization:** Lee A. Armstrong, Sven M. Lange, Stephen P. Matthews, Yogesh Kulathu.

**Formal analysis:** Virginia Dee Cesare, Stephen P. Matthews, Raja Sekhar Nirujogi, Anthony Hope, Fraser Cunningham, Yogesh Kulathu.

**Funding acquisition:** David Gray, Ivan Dikic, Yogesh Kulathu.

**Investigation:** Lee A. Armstrong, Sven M. Lange, Virginia Dee Cesare, Stephen P. Matthews, Raja Sekhar Nirujogi, Isobel Cole, Rachel Toth, Rukmini Mukherjee, Denisa Bojkova.

**Methodology:** Sven M. Lange.

**Project administration:** Paul Davies.

**Resources:** Rachel Toth, Franz Gruber.

**Supervision:** David Gray, Jindrich Cinatl, Ivan Dikic, Yogesh Kulathu.

**Writing – original draft:** Yogesh Kulathu.

**Writing – review & editing:** Lee A. Armstrong, Sven M. Lange, Virginia Dee Cesare, Raja Sekhar Nirujogi, Fraser Cunningham, David Gray, Paul G. Wyatt, Ivan Dikic, Paul Davies, Yogesh Kulathu.

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
