## [Decision Letter · Decision Letter 0]

6 May 2021

PONE-D-21-09090

Characterization of protease activity of Nsp3 from SARS-CoV-2 and its inhibition by nanobodies

PLOS ONE

Dear Dr. Kulathu,

Thank you for submitting your manuscript to PLOS ONE. After careful consideration, we feel that it has merit but does not fully meet PLOS ONE’s publication criteria as it currently stands. Therefore, we invite you to submit a revised version of the manuscript that addresses the points raised during the review process.

The manuscript was reviewed by three experts whose overall comments are positive. They have recommended some specific improvements for revision e.g. the comparison of Nsp3 with PLpro, antiviral potential of nanobodies and more experimental details in figures.

We look forward to receiving your revised manuscript.

Kind regards,

A Ganesan

Academic Editor

PLOS ONE

Journal Requirements:

Reviewers' comments:

Reviewer's Responses to Questions

**Comments to the Author**

1. Is the manuscript technically sound, and do the data support the conclusions?

Reviewer #1: Yes

Reviewer #2: Yes

Reviewer #3: Partly

2. Has the statistical analysis been performed appropriately and rigorously? 

Reviewer #1: Yes

Reviewer #2: Yes

Reviewer #3: I Don't Know

3. Have the authors made all data underlying the findings in their manuscript fully available?

Reviewer #1: Yes

Reviewer #2: Yes

Reviewer #3: Yes

4. Is the manuscript presented in an intelligible fashion and written in standard English?

Reviewer #1: Yes

Reviewer #2: Yes

Reviewer #3: Yes

5. Review Comments to the Author

Reviewer #1: The manuscript by Armstrong et al., provides an in-depth biochemical characterization of the papain-like protease activity (PLpro) within the Nsp3 non-structural protein of SARS CoV-2 towards different substrates (poly-ubiquitin, ISG15 and viral proteins). The authors found that additional domains within Nsp3 are required for the full protease activity against all 3 tested substrates. Despite the observed differences in protease activities, proteomic analysis for the interacting proteins for PLpro and Nsp3 showed a rather similar interacting proteome. They then applied a mass-spectrometry based approach to identify inhibitors for the PLpro activity, using a library of almost 2000 FDA approved compounds. They managed to select 5 inhibitors with relative specificity towards PLpro, which however did not show any anti-viral effects. Finally, the authors using a yeast-based nanobody library they selected nanobodies that bind to the substrate binding site of PLpro and effectively inhibit PLpro activity.

In general, this is a very well performed study and the quality of presented data is excellent. The in-depth biochemical analysis revealed new information on the protease activity within Nsp3, indicating distinct mechanisms for the proteolysis of different substrates. Despite the unfortunate results on the use of the developed inhibitors, I think the used pipeline provides a useful platform for the community for future screens for PLpro inhibitors. The development of nanobodies is clearly a significant advance in the field and a proof of principle for the use of the yeast nanobody libraries to select potent inhibitors.

Based on the comments of the Reviews Common platform, I feel the major issue is to accurately describe the literature on the activity of PLpro and potentially discuss the prospect of the use of the developed nanobodies in vivo. The authors have also provided a satisfactory response on how they will also address any raised technical issues/concerns.

Reviewer #2: Characterization of protease activity of Nsp3 from SARS-CoV-2 and its in vitro inhibition by nanobodies

Manuscript Number: PONE-D-21-09090

Comments:

The ongoing COVID-19 pandemic caused by SARS-CoV-2 has spread worldwide. Effective therapeutic reagents are urgently needed. The Nsp3 with a papain-like protease PLpro domain cleaves the viral protein and polyubiquitin and ISG15 from host cells and, therefore considered as a promising antiviral target. The authors systematically characterized the enzymatic activity of Nsp3 and PLpro of SARS-CoV-2, and found that Nsp3 is more active compared to its PLpro domain, especially for the Nsp1/2 cleavage. Five compounds have been found to inhibit PLpro from a screening of 1971 approved clinical compounds. Further, the authors developed nanobodies that target the active site of PLpro and inhibited the activity of this enzyme. This work provided important information of Nsp3/PLpro catalytic specificity, and demonstrated a promising method for developing anti-PLpro nanobodies.

Major suggestions:

1. Can the authors discuss a bit more on possible reasons that why Nsp3 has more active protease activity compared to PLpro?

2. It is interesting that PLpro is unable to cleave Nsp1-Nsp2, but the full-length Nsp3 can. Can the authors discuss a bit more about the possible reasons?

3. The engineered competitive nanobodies, especially for NbSL18, showed inhibition on PLpro enzyme activity. It would be interesting to show its antiviral activity in future studies.

Minor suggestions:

1. Introduction, “While several coronaviruses express two PLpro enzymes, SARS-CoV-2 only expresses one PLpro enzyme (Barretto et al, 2005).”

Can the authors add another reference after 2020 to show SARS-CoV-2 only has one PLpro?

2. Introduction, “PLproCoV-2 shows many similarities to the PLpro encoded by SARS, PLproSARS.”

Please change “by SARS” to “by SARS-CoV”, as SARS is the name of the disease and SARS-CoV is the name of virus.

3. “Excitingly, our data reveals that NbSL18 can indeed inhibit Nsp1-2 FL cleavage by Nsp3 suggesting that the viral peptide also occupies the same S1 site as ubiquitin (Fig 4H).”

Can the authors add labels to Fig 4H?

Reviewer #3: The manuscript “Characterization of protease activity of Nsp3 from SARS-CoV-2 and its inhibition by nanobodies” by Armstrong et al. compares the enzymatic activity of PLpro vs. Nsp3 towards three substrates; compares the protein interactome of non-membrane-anchored PLpro vs. Nsp3; identifies pharmacological inhibitors of PLpro in vitro; and identifies and characterizes inhibitory nanobodies in vitro.

The amount of data and methodological range is impressive and the data as such seem technically sound. Some of the findings turned out as dead ends, e.g. the inhibitor study which did not yield any candidate that works in cells. However, I agree with the authors’ response to one of the previous reviewers that it is important to report these data.

While there are some interesting findings on which one could have built on (especially the interactome data and the new nanobodies as research tools), the authors decided to limit themselves to discuss existing literature, formulate hypotheses or suggest experiments that could be done. Hence, the manuscript as presented is rather broad and leaves the reader with some loose ends. Still, the data and new reagents might be useful for researchers in the field.

Major comments:

- The promised changes in the answers to the previous reviewers’ comments should be implemented. Particularly fair referencing of existing studies and putting the results in the right context is absolutely essential.

- Page 18/figure 4G: How were the values for ~50% inhibition determined/quantified? By visual inspection of the gel bands for Ub3/2/1, it seems more like 2500nM to me.

- Page 19: The statement “… thus explaining…” suggests certainty. As other possible explanations are possible, this statement should be toned down.

- For the most part it is not clear how often (or whether) experiments were repeated and what the error bars mean (where applicable).

Minor comments:

- Emotional or scientifically uninformative phrasing should be avoided and/or replaced by (semi-) quantitative or specific terms where appropriate (“…very active…”, “To our disappointment…”, “Excitingly…”, “suitable buffer” etc.)

- Page 18: “recently developed” is double

- Page 19: The statement “…current efforts to develop inhibitors are focussed solely on PLpro…” is hard to verify or falsify and should be rephrased (e.g. “so far published studies have focused on…”).

- Page 23: 2YT media?

- Page 25: Under “Qualitative protease assay”: Something is wrong in the first sentence

- Centrifugation speed should be given in xg not rpm

- Some of the figures would benefit from aligning panels, adjusting fonts size and style, etc.

Technical comment:

6. PLOS authors have the option to publish the peer review history of their article (what does this mean?). If published, this will include your full peer review and any attached files.

Reviewer #1: **Yes: **Dimitris Xirodimas

Reviewer #2: No

Reviewer #3: No

---

## [Author Response · Author response to Decision Letter 0]

1 Jun 2021

Separate document with responses to reviewers comments has been attached

---

## [Editor Report · Decision Letter 1]

4 Jun 2021

Biochemical characterization of protease activity of Nsp3 from SARS-CoV-2 and its inhibition by nanobodies

PONE-D-21-09090R1

Dear Dr. Kulathu,

We’re pleased to inform you that your manuscript has been judged scientifically suitable for publication and will be formally accepted for publication once it meets all outstanding technical requirements.

Kind regards,

A Ganesan

Academic Editor

PLOS ONE
---

## [Editor Report · Acceptance letter]

8 Jul 2021

PONE-D-21-09090R1 

­­­Biochemical characterization of protease activity of Nsp3 from SARS-CoV-2 and its inhibition by nanobodies 

Dear Dr. Kulathu:

I'm pleased to inform you that your manuscript has been deemed suitable for publication in PLOS ONE. Congratulations! Your manuscript is now with our production department. 

Kind regards, 

on behalf of

Prof. A Ganesan 

Academic Editor

PLOS ONE